



# Ten years of 1 Hz solar irradiance observations at Cabauw, the Netherlands, with cloud observations, variability classifications, and statistics

Wouter B. Mol[1], Wouter H. Knap[2], and Chiel C. van Heerwaarden[1]

[1]Meteorology and Air Quality Group, Wageningen University, Wageningen, The Netherlands
[2]Royal Netherlands Meteorological Institute, De Bilt, The Netherlands

**Correspondence:** Wouter Mol (wbmol@wur.nl)

**Abstract.** Surface solar irradiance varies on scales down to seconds, of which detailed, long-term observational datasets are rare but in high demand. Here, we present an observational dataset of global, direct, and diffuse solar irradiance sampled at 1 Hz over a period of 10 years, from the Baseline Surface Radiation Network (BSRN) station at Cabauw, the Netherlands. The

dataset is complemented with irradiance variability classifications, clear-sky irradiance and aerosol reanalysis, information about the solar position, observations of clouds and sky type, and wind measurements up to 200 meters above ground level. Statistics of variability derived from all time series include approximately 185,000 detected events of both cloud enhancement and cloud shadows. The Cabauw measurement site has additional observations freely available at the open data platform of the Royal Netherlands Meteorological Institute. This paper describes the observational site, quality control, classification

algorithm with validation, and the processing method of complementary products. These observations and derived statistics provide detailed information to aid research into how clouds and atmospheric composition influence solar irradiance variability, and to help validate models that are starting to resolve variability at higher fidelity. The main datasets are available at https://doi.org/10.5281/zenodo.7093164 (Knap and Mol, 2022) and https://doi.org/10.5281/zenodo.7092058 (Mol et al., 2022b), see the data availability section for the complete list.

## 15  1  Introduction

Clouds generate large intra-day surface solar irradiance variability, the spatiotemporal scales of which reach down to seconds or less (Yordanov et al., 2013; Tabar et al., 2014; Gueymard, 2017; Kivalov and Fitzjarrald, 2018), or tens of meters (Lohmann et al., 2016; Mol et al., 2022a). Observing, understanding, and forecasting irradiance variability at these scales is important for a range of applications. Solar energy production and electricity grid stability is negatively affected by fast and local irradiance

variability (Liang, 2017; Yang et al., 2022). Numerical weather prediction models are incapable of forecasting variability at these short scales, but as their resolution keeps increasing and sub-grid scale irradiance variability parameterizations are developed, they require more detailed observations for validation. Cloud resolving models and the development of more accurate 3D radiative transfer calculations in academic setups (e.g. Jakub and Mayer, 2015; Gristey et al., 2020; Veerman et al., 2022),



likewise require detailed and accurate observations of solar irradiance. Heterogeneity of solar irradiance and resulting surface

fluxes is also an increasingly important topic in the field of land-atmosphere interaction (Helbig et al., 2021), with a non-linear response of vegetation's photosynthesis for varying light intensities (Pearcy and Way, 2012) or diffuse irradiance penetration into canopies (Mercado et al., 2009; Durand et al., 2021).

Existing observational datasets of surface solar irradiance at the sub-minute scale are rare, in particular for multiple years or longer and with separate direct and diffuse irradiance measurements. Notable examples of such datasets include those used

in previously mentioned studies (Tabar et al., 2014; Gueymard, 2017; Kivalov and Fitzjarrald, 2018; Lohmann, 2018; Gristey et al., 2020). In this work, we present such a dataset, which consists of 10 years of 1 Hz resolution global, direct, and diffuse irradiance, supplemented with meteorological observations for interpretation and data analysis. To the best of our knowledge, this is a unique observational dataset given its time span, temporal resolution, and multi-component measurements. The separation of global horizontal irradiance (GHI) into direct and diffuse components is important for distinguishing and characterizing

the different types of atmospheric conditions and specific conditions under which irradiance variability is generated. Most notably, the phenomenon of cloud enhancement, where clouds scatter additional sunlight to cloud-free spots on the surface to significantly increase total irradiance (Gueymard, 2017; Yordanov, 2015; Mol et al., 2022a), is by definition a combination of direct and diffuse irradiance and cannot be understood with only GHI observations.

In this work, we will describe the 10-year observational dataset of solar irradiance, all supplementary meteorological ob-

servations and related processing, the time series variability classification algorithm, statistical datasets derived from all time series and classifications, and examples of how the data can be used. Sections 2 and 3 are a complete and more elaborate version of the condensed dataset description published in Journal of Geophysical Research: Atmospheres (Mol et al., 2022a).

## 2 Observational data description

All in-situ observations in this dataset are taken at the Ruisdael Observatory in Cabauw (previously known as CESAR,

https://ruisdael-observatory.nl/cabauw/) of the Royal Netherlands Meteorological Institute (KNMI). The observatory, hereafter referred to as Cabauw, is located in a rural area in the south west of the Netherlands at 51.97°N, 4.92°E (Figure 1a). The climate in the Netherlands is a typical ocean-influenced west coast climate, with relative mild and wet winters despite its high latitude, and milder summers than further inland. KNMI provides an overview of the current Dutch climate and trends on their website (https://www.knmi.nl/klimaat), including the long term increasing trend of incoming solar radiation and recent

extremes (e.g. that of spring 2020, van Heerwaarden et al., 2021). The following section describes all the observational data we use from Cabauw, the supplementary modeled clear-sky irradiance, atmospheric composition re-analysis, calculated solar positions, satellite-derived cloud types, and ground based cloud cover.

### 2.1 Surface solar irradiance observations

The surface solar irradiance station at Cabauw is part of the Baseline Surface Radiation Network (BSRN, Driemel et al., 2018),

operational since 2005. The BSRN station measures all components of the surface radiation balance. Observations are logged



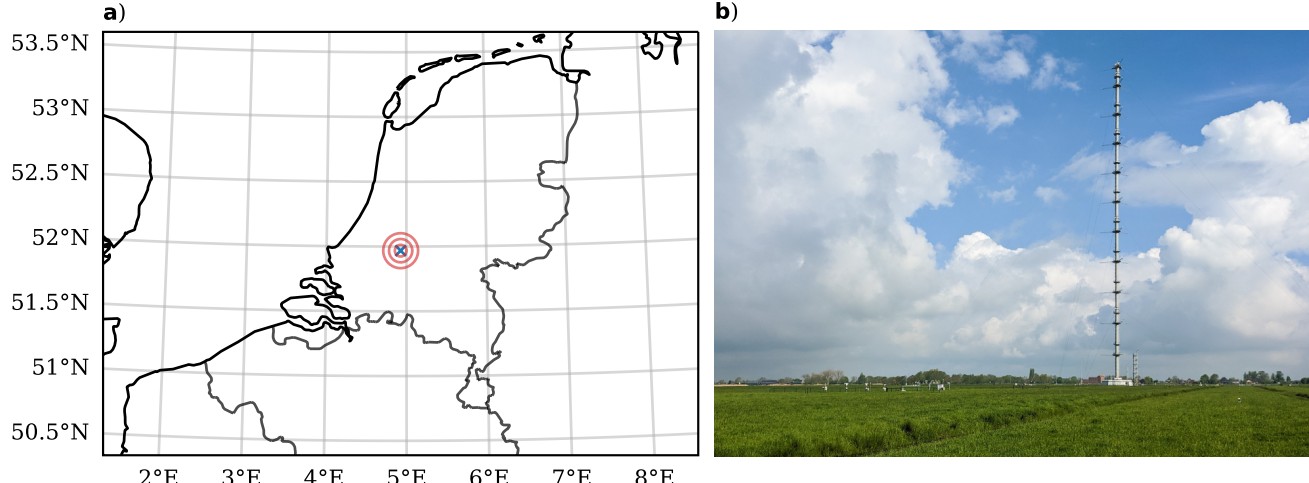

**Figure 1.** Ruisdael Observatory Cabauw, the Netherlands, where all observations in our dataset are located. The geographical location is marked in **(a)** with a cross, the circles are the 5 to 15 km radii for satellite cloud type extraction. A photograph of the 213 meter high tower at the Cabauw site, with the BSRN station among other instruments in the bottom left, is shown in **(b)**.

at a 1 Hz frequency, reprocessed to 1 minute quality controlled and validated data, made available publicly at the PANGAEA data repository (Knap, 2022) with instrument metadata, maintained by station scientist Wouter Knap (KNMI). While 1 minute resolution is enough for many applications, in particular those concerned with the surface net radiation balance at longer time scales, much of the cloud driven irradiance variability occurs at sub-minute scales. For the purpose of research into cloud-

driven irradiance variability at these short time scales, a separate 10-year subset of solar irradiance at 1 Hz resolution has been released (Knap and Mol, 2022), and is described in here. This subset spans from 2011-02 until 2020-12. The three components are global horizontal irradiance (GHI) and diffuse irradiance (DIF), measured with Kipp & Zonen CM22 pyranometers, and direct normal irradiance (DNI), measured with a Kipp & Zonen CH1 pyrheliometer. These instruments are thermopile based, meaning there is a non-zero response time to variations in incoming radiation, and the true resolved resolution is not 1 Hz.

According to manufacturer's specifications, the CH1 pyrheliometer has a 7 second (95%) or 10 second (99%) response time (Kipp & Zonen, 2001), and 1.66 seconds (66%) or 5 seconds (95%) for the CM22 pyranometers (Kipp & Zonen, 2004). Ehrlich and Wendisch (2015) demonstrate a reconstruction technique of the true 1 Hz signal through deconvolution, which is not a trivial exercise. We have not applied this technique, as we cannot validate whether it works reliably for our dataset, but we mention it here as an option to anyone who might want to attempt to apply the method despite its challenges. The

pre-processing we do apply to the data, namely gap filling and quality control, is discussed in Section 3.1.





## 2.2 Supplementary irradiance data

### 2.2.1 Solar position and direct horizontal irradiance

Information about the sun's position is important for quality control, data analysis, and interpretation of results. Calculations of the sun's position (elevation and azimuth angle) are done using the python package PySolar (https://github.com/pingswept/pysolar) at a 1 minute resolution, interpolated to 1 second. The calculations are for the purposes of this research area indistinguishable from highly accurate peer reviewed code such as the Solar Position Algorithm (SPA, https://midcdmz.nrel.gov/spa/). Using the solar elevation angle $\alpha$ (degrees above horizon, or as zenith angle $\theta = 90 - \alpha$), we calculate the horizontal component of direct irradiance: $DHI = DNI \cdot \sin(\alpha)$. An alternative calculation is $DHI = GHI - DIF$, which in the case of a good measurement setup should be equal to $DNI \cdot \sin(\alpha)$, and is the basis for one of the checks in data quality control (discussed in Section 3.1).

### 2.2.2 Clear-sky irradiance and atmospheric composition

Clear-sky global horizontal irradiance ($GHI_{cs}$) is the total downwelling horizontal solar irradiance in the absence of clouds. We use CAMS McClear version 3.5 (Gschwind et al., 2019) as the $GHI_{cs}$ reference for our dataset, released in September 2022. CAMS McClear includes corrections based on atmospheric composition re-analysis such as aerosols and total column atmospheric water vapour. This allows us to define times when GHI exceeds $GHI_{cs}$ as those purely driven by clouds as accurately as possible, as opposed to definitions based on simpler models, which is necessary for the irradiance classification algorithm described in Section 3.3. Atmospheric composition input for McClear is included in their publicly available dataset (https://www.soda-pro.com/web-services/radiation/cams-mcclear), which we add to our dataset for context. The only further processing applied to this data is linear interpolation from 1 minute to 1 second to match the irradiance observations.

## 2.3 Additional in-situ measurements

### 2.3.1 Wind profiles from Cabauw

A 213 meter high tower provides wind speed and direction measurements at 2, 10, 20, 40, 80, 140 and 200 meters above ground level at a 10 minute interval (Wauben et al., 2010). Figure 1b shows the tower with respect to the BSRN site, which is a few hundred meters to the south. We apply no further processing to this data, apart from creating daily files from the monthly files. The original tower data (including temperature, visibility, and humidity) is shared publicly by the KNMI on their open data platform: https://dataplatform.knmi.nl/dataset/cesar-tower-meteo-lb1-t10-v1-2.

### 2.3.2 Nubiscope

Detailed cloud cover observations, for analysis, validation of satellite observations (Section 2.4), and irradiance based sky type classification (Section 3.3.2), we use the observations from a nubiscope located within a few meters from the BSRN instrumentation (Wauben et al., 2010). In 10 minutes, this instruments makes a hemispherical scan of the sky using in-



frared sensors to determine cloud fraction and sky type of various categories. We subset the nubiscope to three years (2014 to 2016) for validation and analyses purposes. If necessary, additional data (2008-05 to 2017-04) is publicly available at https://dataplatform.knmi.nl/dataset/cesar-nubiscope-cldcov-la1-t10-v1-0. Again, no further processing is applied by us, apart from turning monthly files into daily files.

## 2.4 Satellite observations

The satellite product CLAAS2 Benas et al. (2017) provides cloud cover, cloud top pressure (CTP), and cloud optical thickness (COT), available every 15 minutes during day time at an approximate spatial resolution of 20 km$^2$ over Cabauw. We provide three years of satellite data (2014-2016) for both validation and cloud type analyses, and describe the post processing steps in Section 3.2.1.

## 3 Processing and Methods

### 3.1 Quality control and completeness of irradiance data

One of the first steps in the processing is constructing daily files from the raw instrument data, which occasionally misses a few seconds of data. In such cases, we apply linear gap filling between measurement points, after which we apply quality control and derive all other variables. Data quality control for the irradiance measurements is necessary to mask out station maintenance, malfunctioning instruments, or other cases of bad data like those caused by precipitation. Maintenance happens on a nearly daily basis to ensure the high BSRN quality standards are met, and is therefore the most common source of anomalous measurements. It is typically brief and only involves sensor cleaning, though sometimes instrumentation is disabled or replaced due to quality issues such that there are gaps of hours up to a few days. For the official 1-minute BSRN dataset (Knap, 2022), all measurements during such periods are filtered. The 1 Hz version includes quality flags ('good' or 'bad' data) derived from the official dataset, where 'good' means all three components are valid. The 1 Hz version includes the original measurements, and quality flags have to be applied to filter bad data, such that the user can decide on the strictness of filtering themselves. We independently determined data quality at the 1 Hz level by performing the following checks:

1. The absolute rate of change of the DIF and DNI components with respect to clear-sky between two seconds has to be below 5% and 20%, respectively.

2. The same for GHI, except the limit is 5% for cloudy conditions and 20% for sunny conditions. This leads to some false positives, which are reset if GHI and DHI changes are well-correlated.

3. Invalid measurements are padded by 180 seconds before and after to be on the safe side.

4. The residuals $\Delta Q_{abs} = |GHI - (DHI + DIF)|$ and $\Delta Q_{rel} = |GHI / (DHI + DIF) \cdot 100 - 100|$ have to be below 10% and 20 W m$^{-2}$ for a 15 minute time frame, respectively. This time frame is necessary, because the instruments are a few meters apart, which leads to decorrelation of the individual components and larger residuals the shorter the time scale.

5. For 'good' quality, all three components have to pass the tests. If one or more components include missing or bad data, the data for that time if flagged as 'bad'.

The implementation of these rules are in the published processing code (see Section 6). There are only minor differences between these custom 1 Hz based quality flags and the official ones. For all data during during day time (solar elevation angle above 0 degrees), 97.98% of the flags are similar, 1.26% are bad for custom flags and good in official, and 0.77% are good in custom but bad in official. Most of these mismatches originate from just a few days and resulting data is otherwise in very close agreement, with the vast majority of data being of good quality. Figure 2 illustrates for the whole 1 Hz dataset the data availability after (custom) quality control, and shows that most months and years have near 100% complete and good data for all three measured irradiance components. Figure 8 shows that after quality control, DIF + DHI = GHI for every month of the year, as an example data reliability.

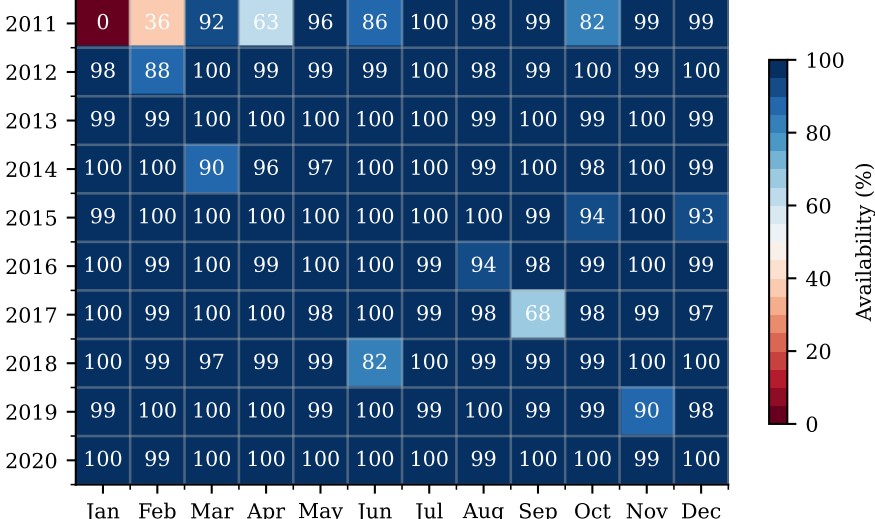

**Figure 2.** BSRN Cabauw 1 Hz data availability per month of available years, during day light (solar elevation angle above 0 degrees), after custom quality control. Numbers are rounded off percentages.

## 3.2 Nubiscope and satellite processing

The nubiscope and satellite data are used both as a validation dataset for irradiance derived sky types (described in Section 3.3), and to provide observations of clouds and sky type for data analysis. Here, we describe the processing applied to the cloud observations, and how the validation dataset is created.



### 3.2.1 Satellite processing

We classify cloud types using a simple cloud top pressure (CTP) and cloud optical thickness (COT) categorization (see NOAA (2022) ISCCP algorithm description, their Figure 20). The 9 types are cumulus (Cu), stratocumulus (Sc), stratus (St) for low clouds; altocumulus (Ac), altostratus (As), and nimbostratus (Ns) for middle clouds; cirrus (Ci), cirrostratus (Cs), and cumulonimbus (Cb) for high clouds. Cu, Ac, and Ci are the optically thinnest clouds for each altitude, and St, Ns, and Cb the thickest, where Cb spans from low to high altitude as the only exception in this list. In this study we use it to group together cloud conditions of various altitudes and optical thicknesses in a more intuitive way, though analyses can be done on the input COT and CTP data rather than derived cloud types. The main limitations are that both the cloud fraction and actual cloud optical thickness contribute to a higher reported optical thickness in a satellite pixel, which is a result of limited spatial resolution, and higher clouds can obscure lower clouds. The spatial satellite product is converted to a time series representative for the BSRN station by determining the most common cloud class within a 5, 10, or 15 km radius around Cabauw, illustrated by the circles in Figure 1a. A smaller radius is not possible due to satellite resolution (pixel area $\approx$ 20 km$^2$) and larger radii become unrepresentative for Cabauw. Cloud cover is derived by calculating the fraction of pixels with clouds within a given radius, which is likely an overestimation due to sub-pixel cloud fractions not always being 1. Overall agreement with the nubiscope is not bad, however, as illustrated by the similar probability densities in Figure 3. Correlation coefficients between the two only show marginal improvement between 10 and 15 km. The satellite derived cloud cover overestimates the extremes at [0.0-0.1] and [0.9-1.0] compared to values bins [0.1-0.2] and [0.8-0.9), and does not have the nuances the nubiscope can resolve. For r = 5 km, there are only 4 pixels, so cloud cover from this is too coarse for most applications, but the dominant cloud type derived from this narrow area around Cabauw is expected to be most representative. This might change for high altitude clouds at low solar elevation angles for example, requiring perhaps more sophisticated approaches, and therefore we have included the original spatial satellite fields in our dataset.

### 3.2.2 Validation dataset

We derive a validation dataset with clear-sky and overcast classifications based on a combination of the nubiscope and satellite data, to be used for statistical verification of sky types based on irradiance observations (Section 3.3.2). Because both instruments have their limitations, the idea is to only identify a situation as clear-sky or overcast if both datasets are in agreement. The rest of the preprocessing involves interpolation to a common 1-minute resolution grid, and masking out any data if either of the two products is missing for given a given time. The validation dataset is provided in Mol et al. (2022b) for all three radii, though we mostly use r = 10 km, for years 2014 until 2016. Disagreement between the two observational datasets is common, with the nubiscope being roughly 3 or 1.5 times more conservative with classifying a sky type as clear or overcast, respectively. This likely has to do not only with the difference in type of observation (ground-based versus remote sensing), but also in that the nubiscope is a more sensitive instrument, as we illustrated in Figure 3 and discussed in Section 3.2.1. In most cases, the nubiscope is more conservative, such that both clear-sky and overcast conditions are mostly controlled by what the nubiscope

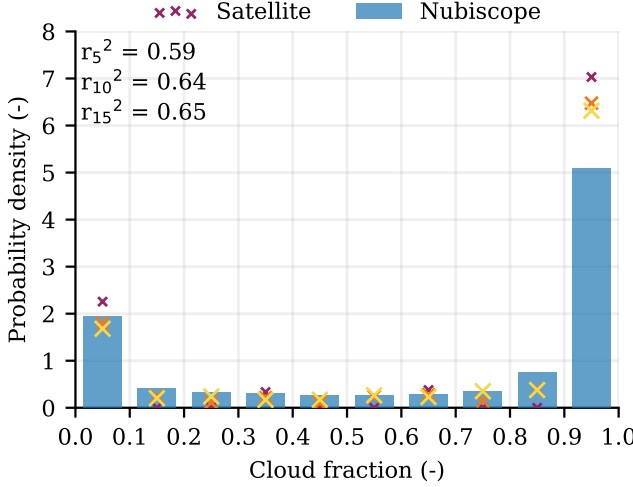

**Figure 3.** Comparison of cloud fraction derived from satellite to the ground-based nubiscope. The analysis is done for radii 5, 10, and 15 km, and show the probability density for 10 bins between 0 and 1 cloud fraction. Satellite radii go from 5 km (dark, small) to 15 km (light, large) cross markers. Correlation coefficients are shown in the top left for each radius. Data ranges from 2014-01 to 2016-12, and is interpolated (nearest-neighbour) to a common 5 minute time axis.

sees. Figure 7 illustrates this best, with the satellite and nubiscope differing in seasonal cycle for clear-sky conditions and in yearly averages for both clear-sky and overcast.

### 3.3 Irradiance classifications

The main addition to the core 1 Hz irradiance time series is the classification of measurements into various categories that describe the type of irradiance variability. We calculate two sets of classification types, one being an instantaneous classification to give a qualification to single measurement point, and the other a more indirect qualification of sky type based on longer time frames. First we describe what the classifications represent and how they are calculated, and then how they are further processed to derive a wide range of interesting statistics about irradiance variability. Examples are shown in Figures 4, 5, and 6, and the
public dataset (Mol et al., 2022b) includes similar quicklooks for all 10 years of time series.

#### 3.3.1 Cloud shadow and enhancement

One of the most noticeable drivers of intra-day irradiance variability are broken cloud fields making patterns of cloud shadows and cloud enhancements (e.g. Yordanov et al., 2015; Gueymard, 2017; Veerman et al., 2022). Cloud shadows are where (most) direct irradiance is blocked, and cloud enhancement where light scattered by clouds locally coincides with direct irradiance
to increase global horizontal irradiance above clear-sky values. We define a shadow where direct normal irradiance (DNI) is below 120 W m$^{-2}$, which is the inverse of what the World Meteorological Organization defines as sunshine (DNI $\geq$ 120 W



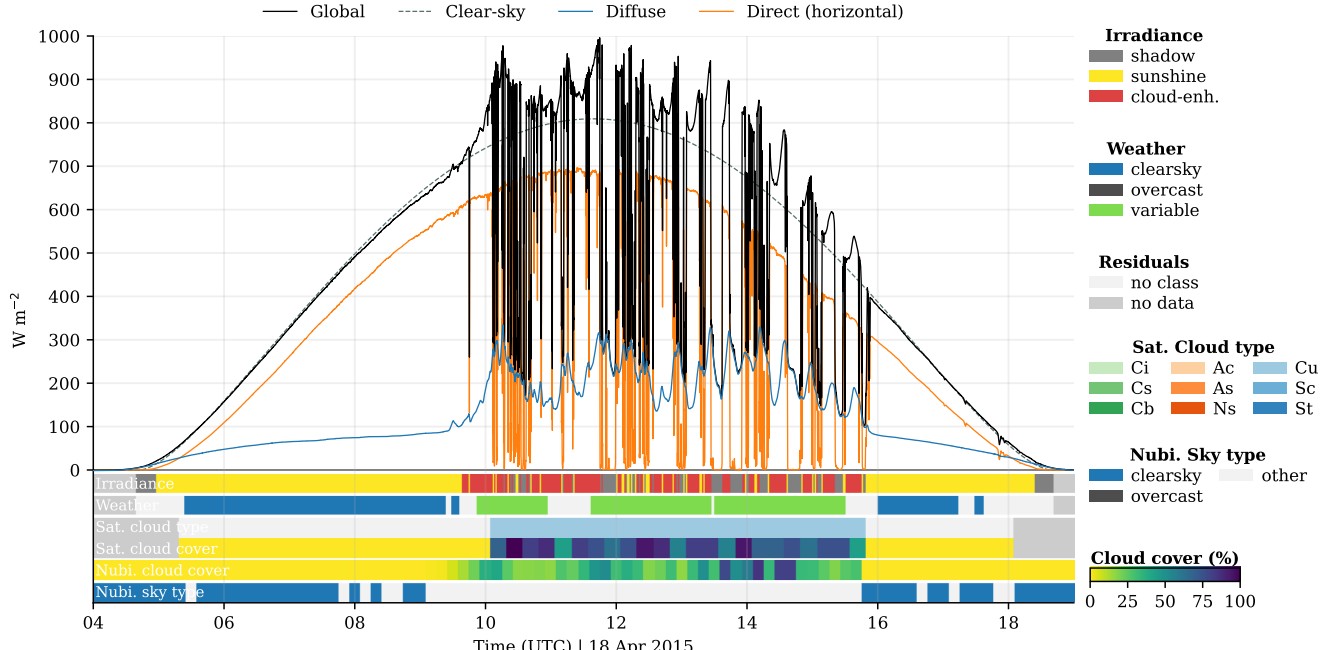

**Figure 4.** Surface solar irradiance time series at 1 Hz, irradiance classifications, and cloud observations for 18 April 2015. It starts with clear-sky conditions and turns into highly variable surface irradiance thanks to scattered boundary layer clouds. The three measured irradiance components, global, diffuse, and direct horizontal irradiance are shown together with modeled (CAMS McClear) clear-sky irradiance.

$m^{-2}$, WMO (2014)), and a straightforward implementation. Cloud enhancement requires a more careful approach. We define cloud enhancement as a single measurement where global horizontal irradiance (GHI) exceeds the reference clear-sky (GHI$_{cs}$). In reality, observed GHI can still fluctuate noticeably in cloud-free conditions, and the GHI$_{cs}$ reference may not be perfect, such that there is some uncertainty in the detection for weak cases of cloud enhancement. To prevent false positives in the detection algorithm, we first apply an activation threshold of GHI exceeding GHI$_{cs}$ by 1% and 10 W m$^{-2}$. Both a relative and absolute threshold are used, as clear-sky irradiance ranges from $10^0$ to $10^3$ W m$^{-2}$ as function of the solar elevation angle (i.e., time of day). 10 W m$^{-2}$ is based on 1% of the typical order of magnitude for clear-sky irradiance around noon for Cabauw. When the threshold is reached, adjacent measurements are also marked as cloud enhancement so long as they exceed GHI$_{cs}$ by 0.1%. Edge cases at low solar elevation angles are removed by requiring DNI to be at least 10 W m$^{-2}$. All of these thresholds are chosen to enable us to capture all but the weakest of cloud enhancements, which arguably are not important. The residual third class is called 'sunshine', which is the WMO definition of sunshine minus cloud enhancement. Detection criteria can be adjusted in the code and recalculated, or another level of filtering can be applied after classification through the derived event statistics (discussed in Section 3.4). Examples of the classifications are shown in the top color-coded bar beneath the time series in Figures 4, 5, and 6.





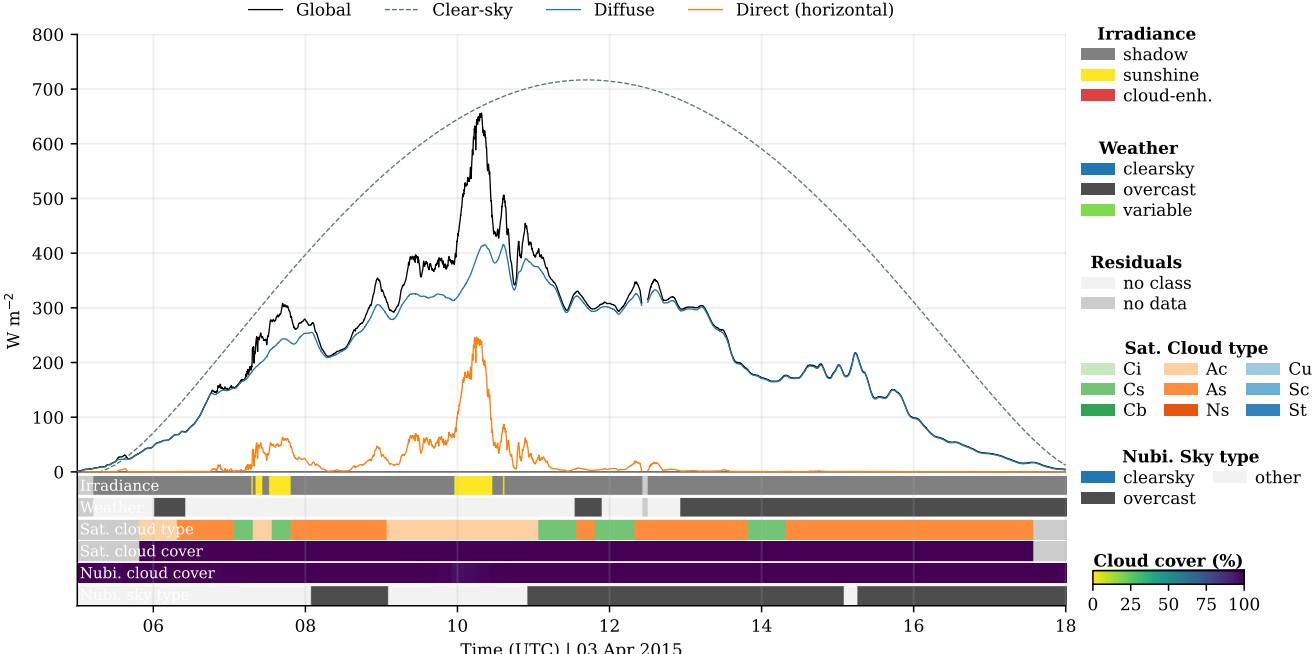

**Figure 5.** Surface solar irradiance observations of a mostly overcast day (3 April 2015), similar layout as Figure 4. The satellite and nubiscope observations indicate overcast conditions of mostly opaque mid to high level clouds (altostratus (As) or cirrostratus (Cs)), though there are brief periods of partially transparent clouds (altocumulus (Ac)).

### 3.3.2 Overcast, clear-sky, and variable irradiance

The second group of classifications represents irradiance 'weather' type, or sky type, based on irradiance data only. The weather types are clear-sky and overcast for smooth and predictable surface irradiance, and a third class of 'variable' irradiance representing pronounced and unpredictable 3D radiative effects to contrast the former two. The way these classifications are derived is partially based on subjective thresholds and assumes good quality clear-sky data, and thus we validate against satellite and ground-based cloud cover observations.

We classify as clear-sky those points in the time series for which, in a 15 minute centered moving window, GHI stays within 3% or 5 W m$^{-2}$ of GHI$_{cs}$ with a maximum standard deviation of the ratio GHI/GHI$_{cs}$ = 0.01 within that window. This irradiance based algorithm emphasizes smoothness, and thus predictability, more than exactly matching GHI$_{cs}$, so as to not rely too much on CAMS McClear being perfectly accurate. Clear-sky conditions are uncommon, occurring between 5 to 15% of the time depending on observational method (Figure 7). Skill scores indicate that the irradiance based classification misses almost half the cases (probability of detection close to 50%), and is generally too conservative (bias < 1) with respect to the validation dataset (see Table 1). The negative bias against the validation dataset, which is not what Figure 7 shows, is due to the skill scores being calculated for cases only when there is agreement between the satellite and nubiscope, rather than for



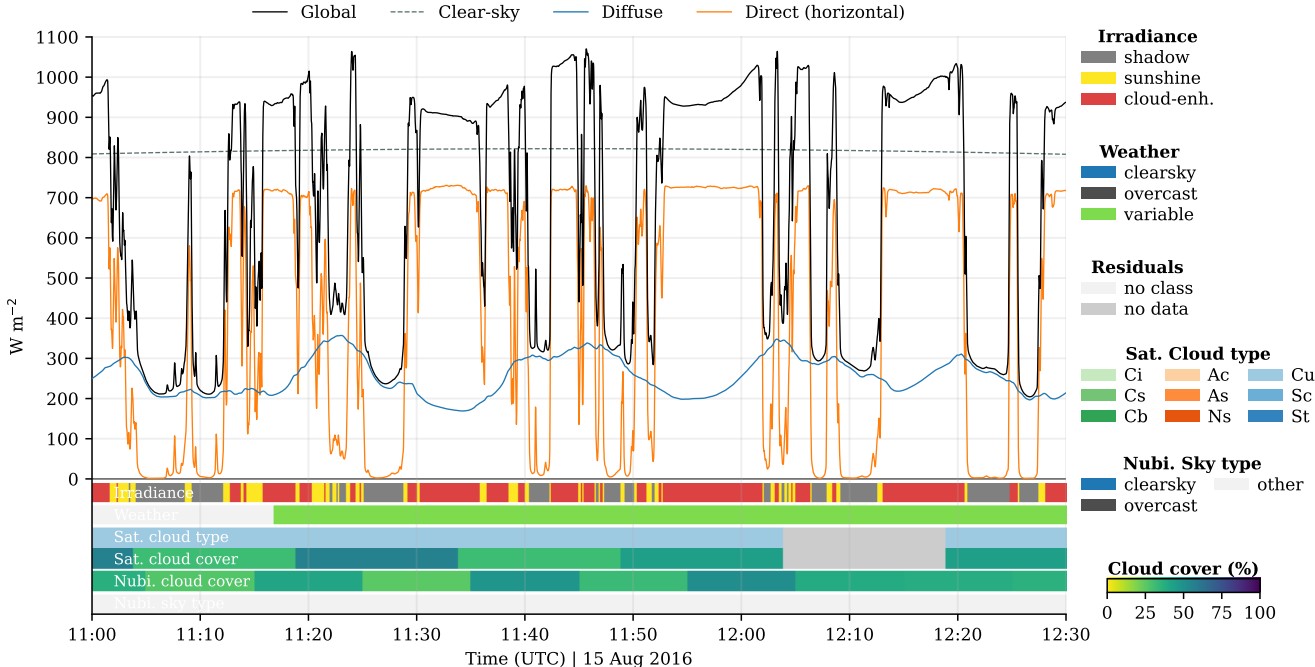

**Figure 6.** Detailed example of cloud-driven irradiance variability. Similar to Figure 4, but for 90 minutes of 15 August 2016.

**Table 1.** Skill scores for irradiance based sky type classifications compared to the validation dataset (satellite + nubiscope). Scores are based on a contingency table approach. POD is Probability Of Detection, FAR means False Alarm Ratio.

| Year | Accuracy | Bias | POD | FAR | Samples (hours) |
|------|----------|------|-----|-----|-----------------|
| **Clear-sky** | | | | | |
| 2014 | 0.969 | 0.624 | 0.478 | 0.233 | 3346 |
| 2015 | 0.962 | 0.610 | 0.494 | 0.191 | 3311 |
| 2016 | 0.968 | 0.666 | 0.486 | 0.271 | 3244 |
| **Overcast** | | | | | |
| 2014 | 0.951 | 0.979 | 0.912 | 0.068 | 2914 |
| 2015 | 0.954 | 0.971 | 0.914 | 0.059 | 2946 |
| 2016 | 0.956 | 0.980 | 0.923 | 0.058 | 2872 |

all available data in Figure 7. The order of magnitude of occurrence is similar to what the validation dataset suggests, but the seasonal cycle is not reproduced, though seasonality between the irradiance classification and satellite alone are similar. For

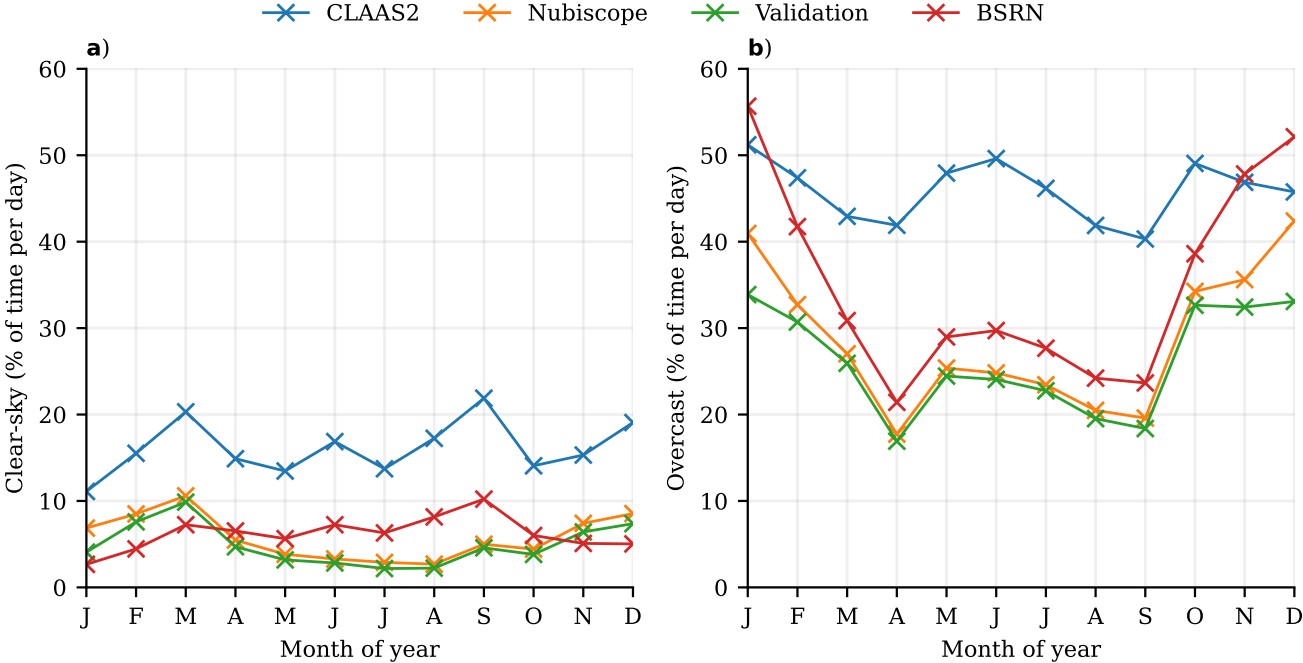

**Figure 7.** Comparison of sky type classifications based on satellite time series (r = 10 km), nubiscope, and 1 Hz irradiance observations. The validation period is from 2014 until 2016, in units relative to available data during daylight (solar elevation angle $\alpha > 0$ degrees).

2014 and 2015, the seasonal correlation to the nubiscope is also much better, but in 2016 there were many cases where the nubiscope saw thin cirrus (cloud cover $< 5\%$, August 17, 18, 23, 24, and 25) rather than clear-sky. In all these cases, GHI $<$ GHI$_{cs}$, consistent with thin cirrus attenuating incoming solar radiation slightly, and not part of the validation set because of disagreement between satellite and nubiscope. It appears (from Table 1 and Figure 9) that there is poor skill in the irradiance-based classification, though manual inspection of quicklooks gives a different impression and most of the bias shown in Figure 9 appears to stem from cases with thin cirrus. If one wants to filter out the thin cirrus cases, the classification threshold can be set more strict and thereby limit cases to more true clear-sky. Figure 5 shows a case with overall agreement between the nubiscope, satellite, and irradiance based clear-sky classification. We refer the reader to the public dataset (Mol et al., 2022b) with time series quicklooks for many more examples.

We define overcast weather as a period of 45 minutes for which the sum of DNI is below 1% of GHI$_{cs}$, and the average below 10 W m$^{-2}$ to catch rare edge cases. This class is indicative of continuous, optically thick and persistent cloud cover, which is a common occurrence in the Netherlands (20 to 50% of the time depending on the season) and does well against the validation dataset. Probability of detection is high (92%), with 6% false alarms and only a slight negative bias of -1.7%. Scores slightly move toward positive (+2.6%) or negative (-5.1%) bias when shortening or lengthening the moving window to 15 minutes and 60 minutes respectively. Although a cloud cover of 100% as seen by the nubiscope and satellite is be classified as



overcast in the validation dataset, cloud cover does not imply the optical thickness is high enough to block all direct irradiance. This distinction is illustrated in Figure 5, which show good agreement for overcast conditions overall, but there is still some irradiance variability with 100% cloud cover around 10 UTC. This example emphasizes our definition of overcast as smooth 240 and predictable diffuse irradiance weather as opposed to a sky type with 100% cloud cover. The seasonal cycle between the irradiance based sky type and nubiscope correlate well (Figure 9), whereas the satellite yearly cycle is less pronounced.

Finally, the variable weather class is defined as any 60-minute window in which 10 transitions from a shadow to cloud enhancement or vice versa occur, built upon the instantaneous classifications defined in Section 3.3.1. It is indicative of weather associated with a characteristic bi-modal distribution of irradiance, cloud enhancements, and at least a handful of large fluc-245 tuations in a short time frame, all of which current numerical weather prediction models cannot reproduce. This classification doesn does well in locating highly variable irradiance conditions, of which examples are shown in Figure 4 and 6, and can for example be used to find case studies.

### 3.4 Event statistics

Within the classified irradiance time series, we call sections of cloud enhancements or shadows 'events'. The 10 years of 250 irradiance time series contain 184,447 cloud shadows and 186,685 cloud enhancement events. For every event, the start and end time are used to select complementary radiation and meteorological data, such that every cloud enhancement and shadow event can be characterized. Notable examples are statistics of event duration, maximum cloud enhancement strength, minimum direct irradiance 'min(DNI)', mean 200 meter wind speed, dominant cloud type, maximum cloud top height, and mean solar elevation angle. Event statistics such as these allows for filtering of events according to additional criteria, e.g. for comparing 255 events of different magnitudes or finding the most extreme cases of cloud enhancement for a given cloud type. Event statistics for cloud shadows and enhancements for are included in the public dataset (Mol et al., 2022b).

### 3.5 Daily statistics

For case study selection or climatological overviews, we calculate daily statistics, which are mostly aggregates of irradiance and classification data. This statistic file is included in the public dataset. We use this to create figures such as Figures 2, 8, and 260 9, and to find specific case studies as detailed in Table 2.

### 4 Examples and use cases

The following section provides some examples and (potential) use cases of the dataset, including previously completed work. Veerman et al. (2022) and Tijhuis et al. (2022) research 3D radiative transfer modelling approaches for cumulus case studies, where 1 Hz irradiance time series and statistics are used as validation. In Mol et al. (2022a), we show how the spatiotemporal 265 scales of cloud shadows and enhancements are described by power laws and driven by cloud size distributions, using the event statistics as described in Section 3.4.

Figures 8 and 9 give an overview of the seasonal and yearly variability of solar irradiance and its classifications that characterize the mid latitude climate of Cabauw. Figure 8 also partially serves are validation of the BSRN instrumentation, with the direct and diffuse components adding up to GHI as should be the case. Figure 9 illustrates the typically overcast conditions
during winter, and highly variable irradiance conditions during summer with significant year to year variability.

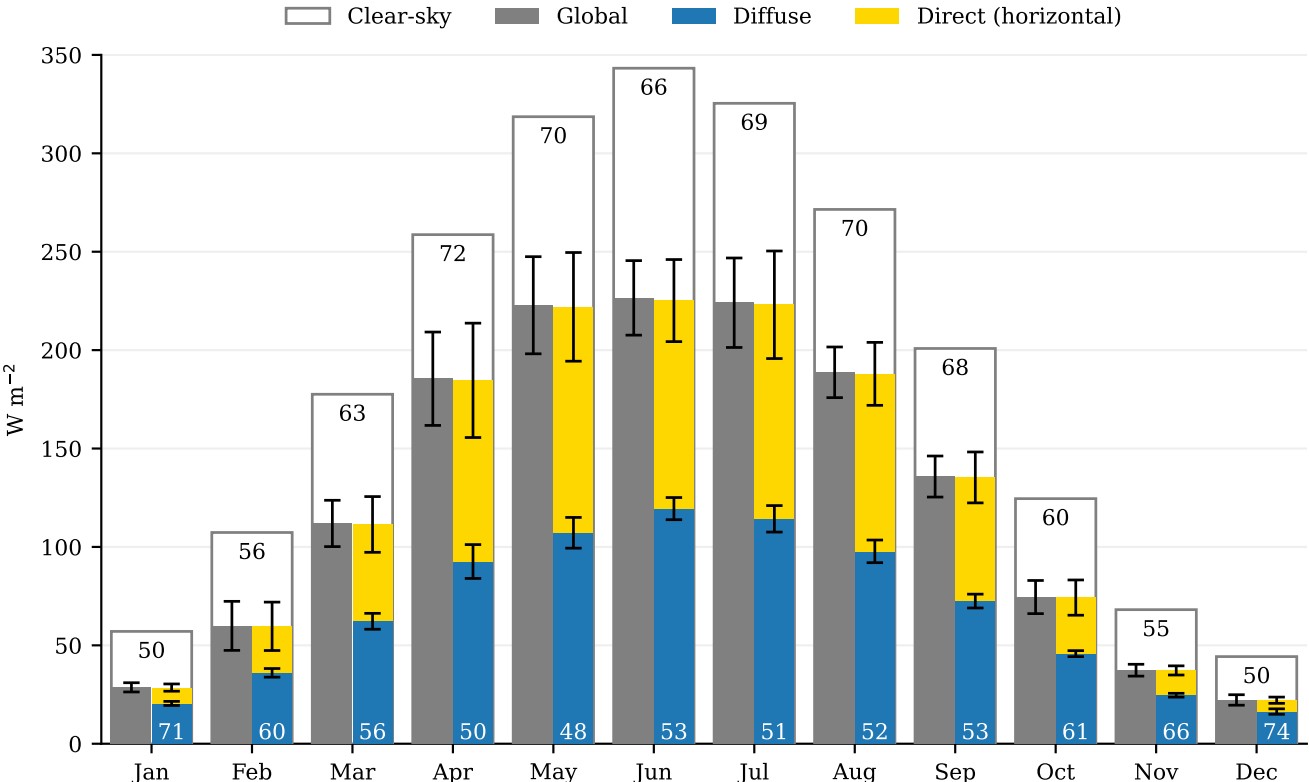

**Figure 8.** Surface irradiance climate for period 2011-02 until 2020-12 based on the 1 Hz dataset of all three components. The numbers in the bottom indicate the percentage of diffuse irradiance (DIF) to global horizontal irradiance (GHI). The error bars indicate, for each component, the year-to-year standard deviation. The white bars encompassing the three components is the clear-sky irradiance ($GHI_{cs}$) for each month, based CAMS McClear, where the numbers in the top are the percentage of GHI compared to $GHI_{cs}$. Only days with $> 95\%$ of data completeness are included.

In order to find specific types of case studies for analysis, you can use either the event statistics or daily statistics to query and filter specific conditions. As an example, we use the daily statistics file and Python's Xarray to find case studies of the most variable irradiance throughout the day or specific cases where overcast conditions transition to clear-sky (or vice versa), which have a potential for brief periods of strong variability and cloud enhancement. These example are shown in Table 2, and
275 the code is publicly available (Section 6).

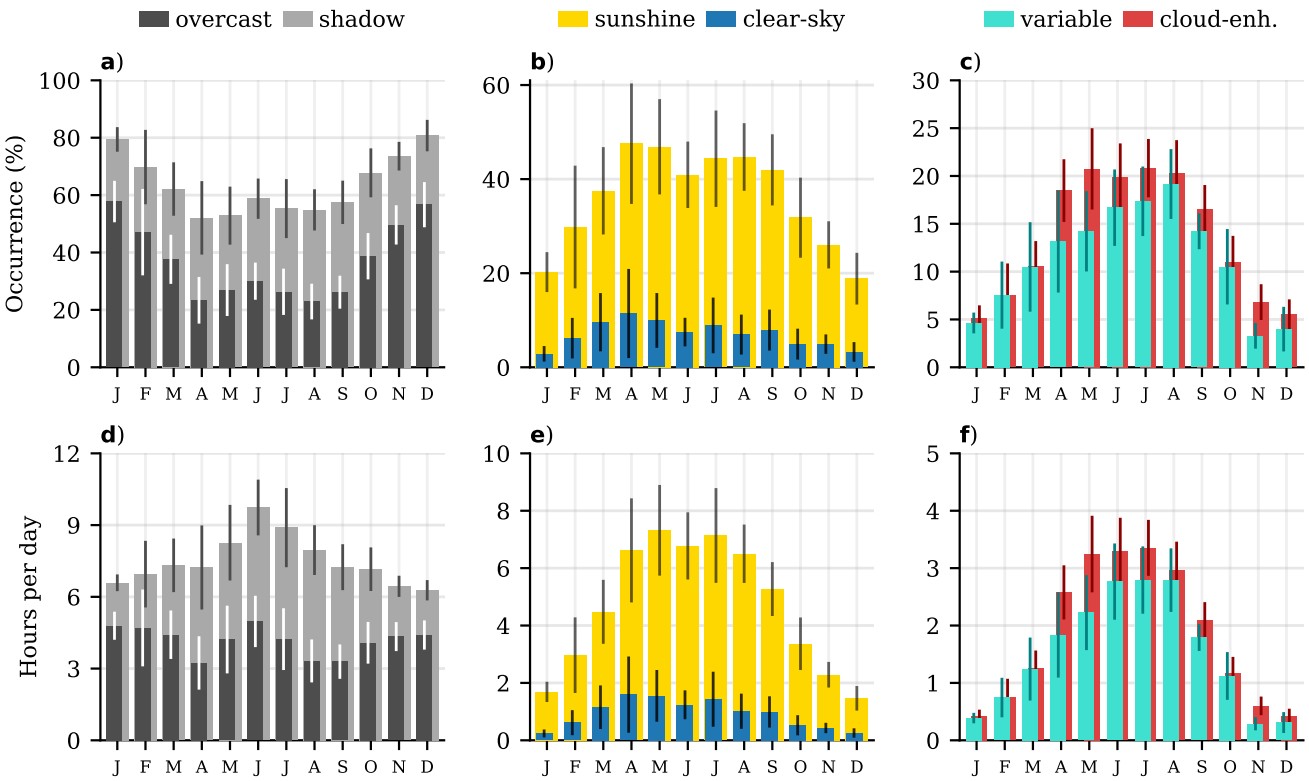

**Figure 9.** Instantaneous and weather classification climatology showing the relative (top row) and absolute (bottom row) occurrence of each classification throughout the year for all available data (2011-02 to 2020-12). Here, sunshine, also includes the portion marked as cloud enhancement, such that shadow + sunshine = 100%. The relative occurrence is expressed as a percentage of daylight (solar elevation angle $> 0°$), absolute occurrence is expressed in average hours per day. Error bars indicate the year-to-year standard deviation.

## 5 Conclusions

In this paper, we describe a high resolution, 10-year long observational dataset of detailed surface solar irradiance, complemented with meteorological data. Using time series classification algorithms, we derive statistics about sky type and irradiance variability. We provide examples and use cases of this dataset to illustrate its potential, ranging from case study selection
and model validation to fundamental insight into drivers of irradiance variability. With all data and processing code publicly available, the user is free to modify our classification algorithm to their liking and validate against independent observations, or expand upon the large set of statistics already provided. Quicklooks for all available days from 2011-02 until 2020-12 are provided to get familiar with the dataset contents and get an impression of the many different types of weather conditions at Cabauw. We believe this dataset is of great use in research into cloud driven irradiance variability, and it provides a necessary
validation reference for models that are starting to resolve the full spectrum of variability.



**Table 2.** Examples of finding case studies using daily statistics. The table shows for a custom period (2014-2016) the top 5 cases for two example queries. The first is the absolute most variable irradiance weather. The second are cases with at least 5% overcast, variable and clear-sky, sorted by those with the most variability, which is a way to find case studies with overcast to clear-sky transitions.

| Top cases | Most variable | Overcast <-> clear-sky |
|---|---|---|
| 1 | 2015-07-22 | 2015-04-30 |
| 2 | 2015-07-17 | 2016-05-17 |
| 3 | 2016-07-02 | 2015-05-12 |
| 4 | 2014-08-09 | 2016-08-06 |
| 5 | 2015-06-18 | 2016-04-19 |

# 6  Code and data availability

1 Hz GHI, DIF and DNI observations of the BSRN station at Cabauw are published on Zenodo (Knap and Mol, 2022), Irradiance time series classifications, supplementary data, quality control, event and daily statistics, and satellite data are published as a separate, complementary dataset on Zenodo (Mol et al., 2022b). Satellite data for an area around Cabauw is taken from the CLAAS2 open access dataset described in Benas et al. (2017), and included in the previous dataset for 2014 to 2016. Also included there is the nubiscope data (Wauben et al., 2010), taken from the KNMI Data Platform (https://dataplatform.knmi.nl/dataset/cesar-nubiscope-cldcov-la1-t10-v1-0) for years 2014 to 2016. All code to reproduce the classifications from the irradiance observations, event and daily statistics, and figures presented in this paper is archived at https://zenodo.org/record/7472545.

*Author contributions.*  WM performed the data analysis, produced the public datasets, and wrote the manuscript. WK is the BSRN Cabauw station scientist and has maintained and produced the original 1 Hz irradiance dataset. CvH has helped shape the manuscript and is project PI. All authors have contributed to the final manuscript.

*Competing interests.*  The authors declare that they have no conflict of interest.

*Acknowledgements.*  WM and CvH acknowledge funding from the Dutch Research Council (NWO) (grant: VI.Vidi.192.068).





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
