# Peer review of "Ten years of 1 Hz solar irradiance observations at Cabauw, the Netherlands, with cloud observations, variability classifications, and statistics"

_Earth System Science Data, 2022_

## Referee Comment (RC1)

**Review of "Ten years of 1 Hz solar irradiance observations at Cabauw, the Netherlands, with cloud observations, variability classifications, and statistics" by Mol et al.**

**2 March 2023**

**Overview**

This paper presents a rich dataset containing an entire decade of surface solar irradiance observations at the Cabauw station in the Netherlands at 1 second temporal resolution. Supporting information on the sky conditions and variability classifications is included that aids interpretation. Several compelling figures are presented that demonstrate the value of the dataset for scientific analyses. The paper is very well written and figures are mostly clear. After addressing the minor comments below I recommend publication.

**Minor comments**

Figure 1: For a reader unfamiliar with this part of the world, it is not immediately clear what is land and what is water. I suggest adding background colors of green for land and blue for water to improve clarity.

L65-66: The 7/5 second response time (95%) for the pyrheliometer/pyranometers is an important caveat given that a central focus of the paper is the 1 second data. While this caveat is already mentioned, I think it deserves more attention. Are there scientific applications where the effective temporal resolution of significantly less than 1 second will be impacted? Is there still a lot of value of data at this response time compared to one-minute data that is already widely available? Please add some discussion on this to the manuscript.

L75: Is linear interpolation used? It will not be exactly linear, but this is probably OK for interpolation within one minute. Either way, best to clarify.

L123-129: Several thresholds are introduced here (5% and 20%, 180 seconds, 10%, 20 W/m2, 15 minutes) without justification. Are these based on trial and error for this study, or an existing method? Please state in the manuscript.

L134: The official BSRN ones? If so, please add "BSRN" here.

Figures 4,5,6: Great visualizations of the dataset. Thank you for also releasing the code to produce these figures. It will be valuable for users to produce these quick looks.

Figure 6: When looking at the timeseries of shadow/sunshine/cloud-enh it seems that the shadows (grey) are always bounded by sunshine (yellow). I expected the opposite: immediately outside the cloud shadows should be the largest enhancement (red). Am I missing something here?

L246: Remove "doesn"

L268: "are" -> "as"

Figure 8: It looks like the DNI+DIF is systematically slightly less than GHI. Is that expected? Could it be related to the measurement, such as the blocking of the direct beam in the pyranometer to get DIF that also blocks a small amount of diffuse radiation that is scattered in the same direction as the direct beam?

Figure 9: I found the legend labels difficult to follow what is actually being plotted. I eventually got there after scanning back through earlier details in the manuscript. I suggest adding some details about each legend label in the caption so that the figure can be interpreted more easily.

---

## Author Comment (AC1)

**Review of "Ten years of 1 Hz solar irradiance observations at Cabauw, the Netherlands, with cloud observations, variability classifications, and statistics" by Mol et al.**

**2 March 2023**

**Overview**

This paper presents a rich dataset containing an entire decade of surface solar irradiance observations at the Cabauw station in the Netherlands at 1 second temporal resolution. Supporting information on the sky conditions and variability classifications is included that aids interpretation. Several compelling figures are presented that demonstrate the value of the dataset for scientific analyses. The paper is very well written and figures are mostly clear. After addressing the minor comments below I recommend publication.

Thank you, we appreciate the kind words. Please see below for our responses to your individual comments (in green text). Our response to the issue relating to true dataset resolution is a combined reply to all reviewers.

**Minor comments**

Figure 1: For a reader unfamiliar with this part of the world, it is not immediately clear what is land and what is water. I suggest adding background colors of green for land and blue for water to improve clarity.

Good point, the map now includes water and land colors.

L65-66: The 7/5 second response time (95%) for the pyrheliometer/pyranometers is an important caveat given that a central focus of the paper is the 1 second data. While this caveat is already mentioned, I think it deserves more attention. Are there scientific applications where the effective temporal resolution of significantly less than 1 second will be impacted? Is there still a lot of value of data at this response time compared to one-minute data that is already widely available? Please add some discussion on this to the manuscript.

Agreed, and both you and the second reviewer have similar concerns regarding our treatment of '1 Hz' in this manuscript. This should have been discussed in more detail. Below, we provide answers to each of your questions, with supporting figures and references, and a list of changes implemented in the revised manuscript.

1. Yes, at 1 Hz, the response time of the pyranometers underestimate the variability and therefore miss part of the physics. By which magnitude exactly we cannot say, given the lack of long term, high-resolution, high-quality observations at a similar location. Our best estimate from power spectra and model comparison is approximately 1 order of magnitude at 1 Hz at worst, or similar at best, while 0.1 Hz (10 sec) is very likely OK. This means this data is not suitable for applications that depend on fully resolved 1 Hz variability.
2. Yes, there is still a lot of benefit from going from 60 seconds to 10 seconds, and even down to 1 Hz there is information. See the power spectra below, which

compares 1 minute to 1 and 0.1 second data. Furthermore, we have successfully used 1 Hz data to derive physical information at scales below 10 seconds: https://agupubs.onlinelibrary.wiley.com/doi/full/10.1029/2022JD037894.

[Figure]

The figure above is the PSD for one year (2016) of 1 Hz data, compared to the same data resampled to 1 minute. The drop-off towards 2 s is partially physical: at some point clouds become so small and thus transparent that variations at these scales are relatively reduced. Cloud shadow to sunlight transitions have a finite amount of sharpness, too. And thirdly, cloud velocity plays a role, where irradiance variability at higher wind speeds will lead to less resolved variability in the signal of our pyranometers, whereas wind-free conditions with slower transitions will be better resolved.

Some caveats here are the limited availability of cloud and irradiance observations or simulations at the scale of 1 Hz for longer time series, especially for the location of this BSRN station.  There is one PSD of a year of data from fast responding sensors (semiconductors) that we know off, located in Hawaii. Below the same figure, with their spectrum overlaid on top (Figure 2 from Tabar et al., https://doi.org/10.1140/epjst/e2014-02217-8):

[Figure]

The relevant line is the red curve, which is the PSD for a single station. While not 1:1 comparable given the different geographical location and climate, we think this is at least some reference of how much our pyranometers might miss at 1 Hz. They show a similar figure based on data from Germany (more comparable location to Cabauw), but unfortunately do not provide information about which instrument was used.

[Figure]

From our own measurements using 10 Hz semiconductor radiometers, we added the spectra for two weeks of data from a field campaign (https://fesstval.de, https://www.fdr.uni-hamburg.de/record/10273), which in contrast to the figure above shows a steeper drop-off between 10 and 1 seconds. The biggest caveat here is the

limited temporal range. Measurements took place during a period with relatively little wind.

Finally, to show that variability is at least resolved at 10 seconds, we refer to Figure 6 of van Stratum et al., 2023 (in review, https://www.authorea.com/users/536402/articles/633946-the-benefits-and-challenges-of-downscaling-a-global-reanalysis-with-doubly-periodic-large-eddy-simulations ), reproduced below. Here, irradiance spectra of semi-realistic large-eddy simulation produces similar variability until 10 seconds (simulation limit). This, in addition to the response time specifications being 10 seconds for 99% of the signal for the slowest responding sensor (pyrheliometer), gives us confidence that all variability is resolved until 10 seconds.

[Figure]

**Figure 6. Solar irradiance spectra:** Power spectra of the surface solar irradiance, comparing ERA5 and LES with the 1 second BSRN observations. The gray shading is the original (not averaged) PSD of the BSRN data.

Given the above, following changes are made to the manuscript:
1. A separate subsection that discusses data resolution, resolved physics, and implications for users.
2. Textual clarifications of the definition of response time and how it relates to sampled vs. resolved resolution.
3. An addition of response time and implications to the abstract.
4. An additional figure of the power density spectrum of the data, which also stresses the significant added value of sub-minute scale resolution. Part of the new subsection.
5. References to work on irradiance spectra, which support its resolved physics until 10 seconds resolution, and added value towards 1 second.
6. Code to produce the spectra figure added to the open dataset with the other scripts

L75: Is linear interpolation used? It will not be exactly linear, but this is probably OK for interpolation within one minute. Either way, best to clarify.

Yes, it is linearly interpolated, which is indeed a good approximation. The text is clarified in the revision.

We double checked the accuracy, and found an RMSE of 0.07 degrees, which translates to a mean absolute error of 0.78 W/m2 and bias of 0.00 W/m2 for direct irradiance of 1000 W/m2 throughout the day, for a theoretical June 1$^{st}$ at Cabauw. This is an extreme case, but I think illustrates that the approximation is well within measurement error (BSRN specifications).

[Figure]

L123-129: Several thresholds are introduced here (5% and 20%, 180 seconds, 10%, 20 W/m2, 15 minutes) without justification. Are these based on trial and error for this study, or an existing method? Please state in the manuscript.

Unfortunately, yes. It was a manual optimization process to get something that works for our use case at 1 Hz. While it may seem arbitrary, it works well enough and only applies to a small percentage of all data. The procedure and code are shared so other users can adjust to make it stricter for example, or simply use official 1 minute quality flags if they wish. The text is clarified accordingly.

L134: The official BSRN ones? If so, please add "BSRN" here.

Indeed, added.

Figures 4,5,6: Great visualizations of the dataset. Thank you for also releasing the code to produce these figures. It will be valuable for users to produce these quick looks.

Thank you!

Figure 6: When looking at the timeseries of shadow/sunshine/cloud-enh it seems that the shadows (grey) are always bounded by sunshine (yellow). I expected the opposite: immediately outside the cloud shadows should be the largest enhancement (red). Am I missing something here?

I believe your reasoning is generally correct for coarser resolution data. However, at 1 Hz we are resolving the variability with such detail that there is always a period (however short) of intermediate sunshine between shadow and cloud enhancement: the transitions phase right at the narrow cloud edges where irradiance is still partially attenuated by the cloud.

L246: Remove "doesn"

Removed, thank you.

L268: "are" -> "as"

Fixed.

Figure 8: It looks like the DNI+DIF is systematically slightly less than GHI. Is that expected? Could it be related to the measurement, such as the blocking of the direct beam in the pyranometer to get DIF that also blocks a small amount of diffuse radiation that is scattered in the same direction as the direct beam?

It is not quite expected, while perhaps it looks more dramatic in the python plot, the closure of the three components is within 0.3 to 0.6%. The cause, however, I do not know. In theory the 'missed' diffuse you mention should be captured by the pyrheliometer, and this is where model and experimental definitions of diffuse/direct irradiance get tricky. Perhaps you will find this paper of interest:
https://www.sciencedirect.com/science/article/pii/S0038092X14004824

It is safe to say that no measurement setup is perfect, though. Luckily, the error here is small and, in our opinion, acceptable, given this dataset's emphasis on variability rather than energy balance.

Figure 9: I found the legend labels difficult to follow what is actually being plotted. I eventually got there after scanning back through earlier details in the manuscript. I suggest adding some details about each legend label in the caption so that the figure can be interpreted more easily.

This has the risk of becoming a very lengthy caption. Instead, I have tried to clarify existing text so that it is clear where it comes from, with a reference to the definitions in Section 3.3.

---

## Author Comment (AC2)

Unfortunately one of the agreed reviewers did not submit their review. In order to move the review process along, I decided in accordance with the journal's policies to provide my own review of the manuscript.

The manuscript presents a unique dataset of 1Hz solar irradiation data combined with additional context such as solar position, satelite cloud cover. The dateset may be useful to test forecast models related to solar energy production, where high frequency variation is important.

Overall, the manuscript is well written and documents the dataset. I have a number of comments that should be included before final publication.

**Thank you for taking the time to review our manuscript as an editor. Please see below for our responses relating to your comments.**

1. True dataset resolution: While data is recorded at 1Hz, the sensor response time is lower. This is unfortunately a technical limitation and dataset users should be aware of this. I would recommend to better highlight the response times and to add response times to the abstract. The authors should also provide a clear definition of response time (i.e. first order response with time constant of ...). Additionally, it would be good if the authors could expand somewhere in the manuscript on the the impacts of the response time on measurements and potential applications.

Agreed, this should have been addressed more in-depth, and a similar point is raised by reviewer 1. Please see my response to their 2nd comment for the full reply to both your concerns regarding response time. In short, we have added a new subsection, figure, and made changes to the abstract to explain the response time and its implications better.

Expand description of data quality flagging. The data quality flagging is important and described very concisely. It might be good to exapand this a bit to make it easier to understand (or to add a flow chart).

We agree on the importance of quality control and flagging of datasets. However, conciseness here does not imply it is not important, and in essence quality flagging is a straightforward process. There are either those 5 listed criteria, or the official BSRN flags from the 1 minute dataset. The resulting availability is then illustrated in figure 2, and the quality of data supported by Figure 8 (closure of DIF+DHI=GHI).

The final sentence of section 3.1 is reformulated to better emphasize the role and interpretation of Figure 8 here. Also, some other parts of section 3.1 have been clarified based on your and reviewer 1's comments.

Maintenance and sensor calibrations: L 115 mentions that sensor/site maintenance happens often. However are there any records on sensor calibrations, maintenance that could be added to the description?

We have specified the exact maintenance schedule now in the text, which is Monday, Wednesday, and Friday. There is no reliable record available for the whole 10 years of data, but indirectly this is mostly captured via data quality control and flags. We agree that ideally, this is interesting metadata to have available, however.

**L 112: Are these gap filled data marked in the quality flags?**

They are not, we now explicitly mention this. It mostly concerns sporadic seconds, and is not at all a widespread occurrence. Should gap filling result in unphysical data, the quality filter would catch those data, but this never occurs as far as we know.

L120: Specify all three components for clarity.

Done.

Figure 4 is very dense. Once could think about ways to make it less so, but it makse sense to overlay the different radiation components. The legend is confusion in te sence that it is not immediately clear to where the residuals map. Also the text on the bars is difficult to read due to white on light grey. Please revise the figures.

It is very dense indeed, but also powerful once the reader/user has figured out the details. I agree with your comments, though, and have made an attempt at improving the clarity of labels and changed some colors.

One naive question about the data in figure 4 is that is appears that cloud enhanced radiation conditions (SWTot>Clear Sky) dominate during cloud activity. If one were to integrat insolation over time, how would this look with respect to total irradiance vs clear sky irradiance

The extremes of irradiance (cloud enhancement and shadows) will cancel out the more temporal or spatial integration/averaging is applied, and should eventually always settle below clear-sky irradiance. Perhaps this is illustrated in an extreme way by the climatology figure (9 in revised manuscript), which shows the average monthly irradiance is well below clear-sky.

Figure 5: I am not sure, I understand this correctly, but according to the text shadow is defined as DNI < 120 W/m2. Around 8 UTC there is data classified as sunshine which does seem to fit the shadow definition from the description text. It would be good if the authors could explain this.

I understand the confusion here, as the definition is for DNI, but for the figure we use the horizontal component DHI. This is so that one can visually add up DHI and diffuse to find the global irradiance, and thereby estimate individual contributions to the sum and validate the setup. Note that this is April 3, at 7 UTC, such that the solar angle is low enough for direct horizontal irradiance to be less than half the direct normal (or 'beam') irradiance.